# Recyclable Composite Membrane of Polydopamine and Graphene Oxide-Modified Polyacrylonitrile for Organic Dye Molecule and Heavy Metal Ion Removal

**DOI:** 10.3390/membranes12100938

**Published:** 2022-09-27

**Authors:** Haoyu Wang, Zhiyun Han, Yanjuan Liu, Maojin Zheng, Zhenbang Liu, Wei Wang, Yingying Fan, Dongxue Han, Li Niu

**Affiliations:** 1Center for Advanced Analytical Science, Guangzhou Key Laboratory of Sensing Materials & Devices, School of Chemistry and Chemical Engineering, Analytical and Testing Center, School of Computer Science and Cyber Engineering, Guangzhou University, Guangzhou 510006, China; 2Guangdong Provincial Key Laboratory of Psychoactive Substances Monitoring and Safety, Anti-Drug Technology Center of Guangdong Province, Guangzhou 510230, China

**Keywords:** wastewater treatment, pH-dependent, membrane

## Abstract

Developing efficient and recyclable membranes for water contaminant removal still remains a challenge in terms of practical applications. Herein, a recyclable membrane constituted of polyacrylonitrile-graphene and oxide-polydopamine was fabricated and demonstrated efficient adsorption capacities with respect to heavy metal ions (62.9 mg g^−1^ of Cu^2+^ ion, CuSO_4_ 50 mg L^−1^) and organic dye molecules (306.7 mg g^−1^ of methylene blue and 339.6 mg g^−1^ of eriochrome black T, MB/EBT 50 mg L^−1^). The polyacrylonitrile fibers provide the skeleton of the membrane, while the graphene oxide and polydopamine endow the membrane with hydrophilicity, which is favorable for the adsorption of pollutants in water. Benefitting from the protonation and deprotonation effects of graphene oxide and polydopamine, the obtained membrane demonstrated promotion of the selective adsorption or desorption of pollutant molecules. This guarantees that the adsorbed pollutant molecules can be desorbed promptly from the membrane through simple pH adjustment, ensuring the reusability of the membrane. After ten adsorption–desorption cycles, the membrane could still maintain a desirable adsorption capacity. In addition, compared with other, similar membranes reported, this composite membrane displays the highest mechanical stability. This work puts forward an alternative strategy for recyclable membrane design and expects to promote the utilization of membrane techniques in practical wastewater treatment.

## 1. Introduction

The rapid development of modern industry has caused serious water pollution, with detrimental impacts on the ecological environment and human health. The numerous kinds of water pollutants can be divided into two categories: organic dye contaminants and heavy metal ions [1,2]. Many technologies have been applied to remove these pollutants from water, for example, photocatalytic degradation/reduction [2,3,4] and electrocatalytic decomposition [5,6]. However, within these catalytic degradation systems, organic dye molecules can only be decomposed from large molecules to small molecular fragments and it is difficult to achieve a complete degradation into carbon dioxide and water. Heavy metal ions can only be converted into less toxic forms. Thus, the pollutant molecules and ions are not actually and entirely removed.

Adsorption filtration has received a great deal of attention with respect to both the removal of organic matter and heavy metal ions [7,8]. Graphene oxide (GO) sheets with numerous hydroxyl groups (-OH) and carboxyl groups (-COOH) possess fantastic hydrophilicity [9,10,11]. The large, conjugated structure of the sheets facilitates the adsorption of dye molecules in aqueous solutions through π-π conjugation [9,10]. Moreover, due to the partial ionization of -OH/-COOH groups, some negatively charged centers (-O^−^/-COO^−^) can be formed and promote the adsorption of positively charged heavy metal ions [12,13]. Therefore, GO sheets display appreciable sewage treatment performance.

There are many adsorption sites on the GO sheets, but π-π stacking and strong van der Waals cause the agglomeration. The hydrophilic properties of GO make it difficult to recycle unless high-speed centrifugation is utilized, and the centrifugal method is unreasonable for the large volumes involved in sewage treatment, such as polluted lakes and rivers. However, weaving GO sheets into a membrane may solve this problem, as it could allow for the maintenance of the malleability of GO sheets such that GO is readily collected. Polyacrylonitrile (PAN) fibers exhibit excellent thermal stability and anti-corrosion properties [14,15]. Thus, the textile membrane was spined with GO sheets and was anticipated to demonstrate a desirable performance in terms of waste elimination in our study.

The reusability of a membrane is another important index to evaluate its performance. This requires the adsorbed pollutant molecules to be desorbed through various experimental treatments. For GO sheets, when immersed in an alkaline solution, the -OH/-COOH groups are deprotonated, forming negatively charged centers (-O^−^/-COO^−^) [16,17] which can push the desorption of negatively charged dye molecules by electrostatic repulsion. However, positively charged dye molecules and cationic heavy metal ions will stick more tightly through electrostatic attraction. Thus, extra positively charged groups are needed to modify the membrane to desorb the dye molecules and heavy metal ions with positive charge. In our system, we would use one of characteristics of polydopamine (PDA) molecules—numerous positive charges amino groups in acidic solutions [18] to create innovative composite materials.

In this paper, composite membranes of PAN/GO/PDA were prepared via an electrospinning technique and an impregnation method. PAN acts as a skeleton in the membranes, guaranteeing the recovery of membranes, and GO/PDA endows the membranes with hydrophilicity and filterability. Taking cationic methylene blue (MB) and anionic eriochrome black T (EBT) as dye molecules and Cu^2+^ as a heavy metal ion, the PAN/GO/PDA membranes exhibited appreciable adsorption performances. In addition, through pH adjustment, the PAN/GO/PDA membranes displayed admirable pollutant-molecule desorption performance, ensuring antifouling abilities and reusability. Equally important, compared with other reported electrospinning films, the PAN/GO/PDA membranes possessed higher mechanical properties, such as tensile strength and mechanical-damage resistance. This work is anticipated to promote the development of membrane technology in wastewater treatment.

## 2. Materials and Methods

### 2.1. Materials

All the materials, including dopamine hydrochloride (DA, 98%), graphite, tris(hydroxymethyl)-aminomethane (Tris, ≥99%), and polyacrylonitrile (PAN, Mw 150,000) were used as-received from Aladdin Chemical Co., Ltd. (Shanghai, China). All aqueous solutions were prepared with doubly distilled water (DI water) from a Millipore system (Chengdu, China) (>18 MΩ cm).

### 2.2. Preparation of GO

GO was prepared according to the modified Hummer’s method [19]. Briefly, graphite powder (0.064 g) was poured into 100 mL of concentrated H_2_SO_4_ in an ice-water bath. Then, 1 g NaNO_3_ and 6 g KMnO_4_ were gradually added. The mixture was stirred for 2 h, then diluted with distilled water. After that, 10 mL 30% H_2_O_2_ was introduced to the solution until the mixture turned brilliant yellow. Then, the graphite oxide was exfoliated with ultrasonic treatment after the suspension was poured into DI water. The mixture was dialyzed with DI water to remove acids and metal ions. After dialysis, the GO solution with a calibrated concentration was dropped into proper ethanol under ultrasonic treatment and vigorously stirred to prepare the GO ethanol solution.

### 2.3. Preparation of the PAN Membranes and the PAN/GO Nanofiber Membranes

All the nanofiber membranes were prepared by electrospinning. All the membranes were fabricated with a solution of PAN (12%, DMF). The raw PAN membranes were fabricated in a spinning environment (at ambient temperature and a humidity of 45 ± 5%) using a home-made electrospinning machine. The electrospinning process was operated with a feed rate of 0.9 mL h^−1^, an applied voltage of 18 kV, and a distance between the spinneret and the roller of 12 cm.

The fabrication process for the PAN/GO membranes is shown in Figure 1a. The operation of the electrospinning part was the same as for the single-component PAN membranes, while a GO solution (0.3 mg mL^−1^, ethanol) was scattered onto the other side of the roller collector by ultrasonic spraying with a wide field at 20 watts of power and with a 10 cm distance from the sprayer nozzle to the collector. To improve the controllability of composition, the width of the collector was controlled at 5 cm. During the whole process, the collector was heated by an infrared lamp from the beginning to the end to evaporate the solution. The feed rates for the GO solution were 0.25 mL min^−1^, 0.5 mL min^−1^, 1.0 mL min^−1^, and 2.0 mL min^−1^, and the inlet nitrogen flow rate was 10 L min^−1^. The prepared samples were 0.25 PAN/GO, 0.5 PAN/GO, 1.0 PAN/GO, and 2.0 PAN/GO.

### 2.4. Preparations of the PAN/PDA and PAN/GO/PDA Composite Membranes

The PAN/PDA and PAN/GO/PDA composite membranes were prepared based on the polymerization of DA on the surfaces of PAN nanofibers and GO sheets, respectively. Typically, PAN/GO membranes with different material proportion ratios were dipped into a solution composed of DA (0.8 g L^−1^) and tris (1.2 g L^−1^) buffer at pH = 8.5, under stirring for 12 h at room temperature. After that, the membranes were taken out and flushed with deionized water 3 times by suction filtration and dried by lyophilization for the subsequent tests. The PAN/PDA membranes, in contrast, were obtained from the PAN membranes via the same treatment processes used in the fabrication of the PAN/GO membranes.

### 2.5. Characterization

The prepared membranes were characterized by Fourier reflective infrared (FTIR; TENSOR II+ Hyperion 2000, Shanghai, China) and X-ray photoelectron spectroscopy (XPS; Escalab 250xi, Billerica, MA, USA), as well as atomic force microscopy (AFM; Agilent 5500AFM, Santa Clara, CA, USA). The structures and morphologies of all the composite membranes were characterized by scanning electron microscopy (SEM; XL30ESEM-FEG, Hillsboro, OR, USA). The water contact angles of the membrane interfaces were tested with a contact-angle testing instrument (ZHIJIA ZJ-6900, Shenzhen, China).

### 2.6. Adsorption Capacity of Ionic Dyes of the Composite Membranes

Ionic dyes, as candidates, were dissolved in deionized water at a concentration of 50 mg/L to evaluate the adsorption capacities of all the membranes. The permeability fluxes of the as-prepared membranes were characterized by means of a dead-end flow filtration experimental device connected to a solution reservoir (core plate diameter: 43 mm). Before testing, the as-prepared membranes were spread out and fixed on the sand core and then a proper dye solution was poured into the reservoir until the liquid level reached 10 cm. Accordingly, the membranes were soaked and the filtrate dropped down. In the meantime, to sustain the fluid level, a dye solution was injected into the reservoir with a peristaltic pump. The liquid flux of the membranes was calculated according to the following formula:$$J=\frac{V}{A\;t}$$
where *J* represents the flux of the membranes (mL cm^−2^ h^−1^) and *V*, *A*, and *t* represent the volume of the permeation solution (mL), the effective area of the membranes (cm^2^), and the permeation time (h), respectively.

Organic MB and EBT molecules were selected as the candidate dyes to represent cationic dyes and anionic dyes, respectively, in order to estimate the adsorption capacities of the composite membranes.

### 2.7. Reusability of the Membranes

The reusability of the composite membranes was also investigated using MB and EBT solutions at concentrations of 50 mg L^−1^, according to a method similar to the one described above. A dye solution was filtered through a composite membrane of 20 μm thickness and the processes were carried out under the same conditions as aforementioned. A dye solution was filtered through the composite membrane, while the percolate was investigated by ultraviolet spectroscopy to determine the concentration. Until the concentration of percolate was the same as that of the raw dye solution, the saturated adsorption capacity was calculated by applying the following formula:*C_a_* = (*c_d_*−*c_p_*) *V*
*M*/*m*
where *c_d_*, *c_p_*, *V*, *M*, and *m* are the concentrations of the dye solution and the percolate, the solution volume, the molecular mass of the dye, and the mass of the membrane, respectively.

To elute the EBT adsorbed, the membranes were flushed with appropriate amounts of NaOH solution (pH = 10.2) until the percolate became colorless. Then, the membranes were flushed with a certain amount of ethanol and water instead. After repeating the above procedures twice, the EBT adsorbed in the membranes was mostly removed. The adsorption capability of the recycled composite membranes for EBT was tested for more runs in a similar operation to the one described above.

The elution of MB in the membranes was similar to that of EBT. The membranes were flushed with diluted HCl solution (pH = 1) until the percolate became colorless. After that, the membranes were flushed with a proper amount of ethanol instead. Most of the dye was removed after repeating the above operations twice. Similarly, the adsorption capacities of the recycled composite membranes for MB were tested for more runs according to the same method used for EBT.

### 2.8. Experiment Testing Adsorption of Cu^2+^

The adsorption capacities of the composite membrane for Cu^2+^ were also evaluated according to a method similar to that used to evaluate the adsorption of ionic dyes. First, a certain amount of CuSO_4_ was dissolved with deionized water to prepare the solution (50 mg L^−1^). A membrane sample was fixed between a dead-end flow filtration experimental device and a solution reservoir (core plate diameter: 43 mm). Before testing, the as-prepared membrane was spread out and fixed on the sand core and then the CuSO_4_ solution prepared was poured into the reservoir until the liquid level reached 10 cm. Accordingly, the membrane was soaked and the filtrate dropped down drop by drop. To ensure saturated adsorption, the filtrate was pumped back into the reservoir cyclically for 60 min and the liquid level in the reservoir was kept constant. The Cu^2+^ concentration of the filtrate was determined by inductively coupled plasma–optical emission spectrometry (ICP-OES, Billerica, MA, USA). The saturated adsorption capacity was calculated by applying the following formula:*
C_a_ = (c_d−_c_p_) V M/m
*
where *c_d_*, *c_p_*, *V*, *M*, and *m* are the concentrations of the Cu^2+^ solution and the percolate, the solution volume, the molecular mass of Cu^2+^, and the mass of the membranes, respectively.

### 2.9. Mechanical Strength Test of the Membranes

Tensile strength and elongation Young’s modulus were measured using a model UTM2203 electronic universal testing machine (Jinan Huike Test Instrument Co., Ltd., Jinan, China) at room temperature at a constant crosshead speed of 5 mm min^−1^, with an aluminum sample holder.

## 3. Results

### 3.1. Synthesis and Characterizations

The target PAN/GO/PDA membranes were fabricated via an electrospinning technique and subsequent polymerization (Figure 1a). The PAN/GO composite was first prepared by the electrospinning technique, according to which the GO solution was sprayed with a nitrogen gas flow onto a roller collector, while PAN was spined on the other side under suitable parameters. During this process, PAN nanofibers and larger GO sheets were stacked together crosswise. The nanofibers were rapidly sprayed from the nozzle to the collector at a high voltage, with rapid rotation of the roller collector, so that they were piled on the collector layer by layer as non-woven fabrics. An ethanol solution of graphene oxide was nebulized with an ultrasonic nebulizer and formed a jet stream by carrier gas (N_2_). When the nozzle and the ultrasonic nebulizer were separated on both sides of the roller collector, electrospinning and ultrasonic spraying were performed simultaneously, with the collector rolling at a certain speed. Along with the jet stream, microbeads of the GO solution stuck to the fibers, and the GO sheets were left adhering to the fibers with the evaporation of ethanol. Therefore, as the collector rotated, the GO sheets and nanofibers would stack layer by layer. Thus, the nonwoven PAN fibers were used as a support and the GO sheets were tiled and anchored onto the fibers in the membranes. Then, the synthesized PAN/GO composites were dipped into dopamine solution. Due to the abundant oxygen-containing functional groups and the π-conjugated structure on the surface of the GO, the dopamine was largely adsorbed and polymerized into PDA. Finally, satisfyingly hydrophilic PAN/GO/PDA membranes were prepared.

The successful fabrication of the PAN/GO/PDA, PAN/GO, and PAN membranes was verified by attenuated total reflection–Fourier transform infrared spectroscopy (ATR-FTIR; Figure 1b). The reflectance peaks of C-H (2935, 1453, and 1373 cm^−1^) and C≡N (2243 cm^−1^) groups confirmed the existence of PAN nanofibers in both the PAN/GO/PDA and PAN/GO samples [20,21]. The incorporation of GO can be verified by the reflectance peaks from 3100 to 3600 cm^−1^, representing -OH groups [22,23]. The successful polymerization of dopamine in the membranes was reflected in the N-H peaks (1573 cm^−1^) for PDA [24,25]. The XPS survey spectra could also provide proof of the favorable recombination among PAN, GO, and PDA. As shown in Figure 1c, compared with GO, a distinct N1s peak appeared at 400.5 eV, indicating integration between GO and PAN [26,27]. Subsequently, the successful incorporation of PDA into PAN/GO was further reflected in the enhanced N1s peak (Appendix A), since the PDA molecules also contain large numbers of N atoms. Magnifying the N1s peak of PAN/GO/PDA at high resolution shows two peaks at 399.8 eV (C≡N) and 400.6 eV (pyrrolic N). C≡N and pyrrolic N are the characteristic peaks for PAN and PDA, respectively [28,29], signifying the combination between PAN and PDA in the PAN/GO/PDA membranes. Additionally, the peaks of C=O (288.7 eV), C-O (286.6 eV), and C=C/C-C (284.8 eV) from the high-resolution C1s XPS spectra confirmed the coverage of GO in the PAN/GO/PDA membranes [30,31], and the C-N peak (285.7 eV) verified the existence of PDA in the PAN/GO/PDA membranes [32].

The construction and morphology of the PAN/GO/PDA membranes were monitored by SEM and AFM. The proportion of GO in the membranes was determined by the flow rate of the GO solution, and the variation could affect the morphology and structure. During the spinning process, the deposition rate of PAN nanofibers was constant because the spinning parameters were not changed, while the deposition speed of the GO solution spray was determined by its flow rate. At the same time, the higher the flow rate, the higher the density of GO microdroplets attached to a unit area. Therefore, as the flow rate of the GO solution increased, the concentration of GO that could be deposited on each layer of fibers increased proportionally [20,33,34]. As shown in Figure 2a, the PAN fibers with an average diameter of 0.3 μm crosslinked perfectly with each other, establishing the frameworks of the membranes, whereas the loose pore structure with ca. 2 μm diameter was unfavorable for filtering nanoparticles and small molecules. To increase filtration interception resistance, single-layer GO sheets were used to patch the large pores in the PAN membranes. The AFM image shows that the average diameter of the GO sheets was 2 μm, exactly matching the apertures of the PAN membranes. In addition, the GO sheets tested had single-layer structures and thicknesses of ca. 1.1 nm (Figure 2c), facilitating the stretching of GO sheets between the adjacent PAN fibers and the covering of the large pores of the PAN fibers. The coverage of GO is determined by the spraying rate. When the spraying speed of the GO solution was relatively low (0.25 mL min^−1^), the GO layers exhibited a lesser distribution in the composite membranes (0.25 PAN/GO; Appendix A) and most pores were uncovered. As the spraying speed increased to 0.5 mL min^−1^ (0.5 PAN/GO; Appendix A) and 1.0 mL min^−1^ (1.0 PAN/GO; Appendix A), the proportion of GO in the composite membranes obviously increased and the pores were progressively covered. Until the speed increased to 2.0 mL min^−1^, almost all of the pores and fibers were covered (1.0 PAN/GO; Figure 2d and Appendix A), while many tiny pores and residual holes in the GO sheets were still left, which were caused by evaporation, so the permeability of the composite membranes could be retained. In order to make sure that PDA could anchor onto the PAN/GO membranes uniformly, the polymerization reactions took place under high-speed stirring in an open system to ensure a sufficient oxygen atmosphere for polymerization. Finally, as shown as Figure 2e, the PDA was uniformly dispersed on the surface of the 2.0 PAN/GO membranes without structural destruction.

The cross-sectional morphology of the 2.0 PAN/GO/PDA membrane was monitored to reveal the transverse channel path. The appearance of the 2.0 PAN/GO/PDA membrane is a thin-film structure (Figure 3a), with a thickness of ca. 16 μm (Figure 3b). Magnifying the cross-section revealed that a large number of pore channels exist among the PAN fibers (Figure 3c,d). These pore channels facilitate the flow of penetrating solution between two layers of GO/PDA and guarantee the interception of small molecules in the membrane. Furthermore, the BET surface area and porosity of 2.0 PAN/GO/PDA were tested to support the conclusion regarding porosity properties (Appendix A).

### 3.2. Hydrophilic Tests of the Membranes

A water-pollutant adsorption film must possess satisfactory hydrophilicity. Thus, the hydrophilicities of the membranes were tested through water-contact-angle measurements. For raw PAN membranes, a poor hydrophilicity was demonstrated, with a water contact angle of more than 66° (Figure 4a) but a large water flux due to the large pore structure (20.8 mL cm^−2^ h^−1^; Figure 4b). This property is unfavorable for the adsorption of pollutants in aqueous solution, since sewage can only pass through the channel fleetingly, without contact with the membrane. After the modification of GO/PDA, membrane hydrophilicity was enhanced, with water contact angles descending to 43° (2.0 PAN/GO/PDA). Additionally, with the blocking effect of GO/PDA, the water flux was reduced to 1.4 mL cm^−2^ h^−1^ for the 2.0 PAN/GO/PDA membrane, enabling persistent and sufficient contact between sewage and membrane, which could promote the absorption efficiency of pollutant molecules on the surface of the membrane. It should be noted that characterizing the initial water contact angle of the membrane is difficult because of the permeability; thus, values at 0.5 s were acquired as comparisons (Appendix A). Among these results, the result for 0.25 PAN/GO/PDA was atypical. This result could be attributed to the proportion of GO and the structure of the membrane. In the process of the ultrasonic nebulization of the GO solution, the fragmentation of GO sheets was more obvious when the liquid flow rate was lower, so a large number of GO sheets of small scales would wrap on the fiber surface without sealing the holes between fibers. The wrappage on fibers formed a lot of wrinkles that increased the roughness and capillary action. Hence, water would permeate rapidly, and the contact angle would decrease rapidly with time. As the flow rate of nebulization increased during membrane preparation, more GO sheets with large scales appeared in the membranes (0.5 PAN/GO, 1.0 PAN/GO, and 2.0 PAN/GO), and more holes between fibers were sealed. The surfaces of fibers in these membranes were smoother, and the layered-distribution structure of GO was more obvious. Therefore, the water would tend to spread out along the GO sheets rather than percolating down. In general, the surface roughness and penetration rate may be the key factors in the atypical behavior of the 0.25 PAN/GO/PDA sample.

### 3.3. Adsorption Performance Evaluation

The adsorption performances of the PAN/GO/PDA membranes with respect to heavy metal ions and organic dye molecules have been investigated using Cu^2+^ ions and cationic MB and anionic EBT dye molecules as candidates. As shown in Figure 5a–c, the raw PAN membrane demonstrated poor adsorption capacity for all Cu^2+^ ions and MB and EBT dye molecules. This should be attributed to the inadequate hydrophilicity, large pore structure, and drastic water flux of the PAN membrane, which is not conducive to the retention and adsorption of pollutant molecules on the membrane surface. After the introduction of PDA, adsorption improvements with respect to Cu^2+^ ions and MB and EBT molecules were observed for PAN/PDA in the initial conditions without acidification or alkalinization, because of the increased hydrophilicity and π-conjugated construction of the PDA. These features are favorable for ion contact in aqueous solution and adsorption by π-π conjugation with MB and EBT [35,36]. As mentioned above, the engagement of GO further promotes the hydrophilicity of the membranes and increases the retention time of water through lessening of the water flux. As expected, along with the engagement of GO, the Cu^2+^, MB and EBT adsorption capacities were boosted, the maximum capacities achieved with 2.0 PAN/GO/PDA. However, the excessive spinning of GO leads to decline in Cu^2+^, MB and EBT adsorption. This is attributed to the fact that excess GO could stack together and seal up the pores in the membrane and obstruct the flow. The adsorption process, along with the time allowed for the adsorption of the dyes, can be divided into two stages (pH = 7; Figure 5d): stage 1, the reaction time from 0 to 4 h; and stage 2, from 4 to 12 h. In stage 1, the adsorption capacity dramatically increases in a short time, indicating a high adsorption rate of the membrane toward organic dyes, while in stage 2, the increase in adsorption capacity becomes sluggish and the adsorption rate is dramatically decreased. The high adsorption rate in the initial stage (1) is attributed to the high concentration of dye in the solution, while the low adsorption rate in the latter stage (2) is related to the saturation effect and the greatly reduced dye concentration in the solution [37,38].

The pH-dependent adsorption and desorption properties of PAN/GO/PDA have been examined in relation to organic pollutant molecules. Huge differences in the adsorption performances of PAN/GO/PDA were observed for MB and EBT. Compared with neutral solutions, the PAN/GO/PDA composite membrane displayed a ca. 4.7-fold adsorption improvement for cationic MB in alkaline solutions (pH = 10) and a ca. 15.1-fold reduction in acidic solutions (pH = 1). On the contrary, for anionic EBT, the adsorption capacity was improved by 3.8 times in acidic solution, yet it declined by 7.8 times in alkaline solution. This indicates that the adsorption and desorption behavior of the PAN/GO/PDA membrane with respect to dye molecules can be controlled by adjusting pH values. Taking 2.0 PAN/GO/PDA as an example, the dependence of adsorption capacity on pH value was investigated by adjusting pH values from 1 to 10. As shown in Figure 5e, the adsorption capacity of MB was positively correlated with pH value, where the adsorption capacity was the smallest (4.1 mg g^−1^) in the acidic solution (pH = 1) and significantly increased to 306.7 mg g^−1^ with pH = 10. An inverse variation trend was observed for EBT adsorption. The 2.0 PAN/GO/PDA membrane displayed the highest adsorption capacity (339.6 mg g^−1^) in acidic conditions (pH = 1) and the lowest in alkaline conditions (11.3 mg g^−1^, pH = 10). The effect of pH value on adsorption–desorption behavior towards MB and EBT is mainly ascribed to the protonation and deprotonation effects of PDA in the PAN/GO/PDA samples, which subject is elaborated on in the section on mechanisms. The isothermal adsorption experiments were carried out and the details are listed in the Appendix A (Appendix A). Based on the results, the composite 2.0 PAN/GO/PDA membrane showed the maximum adsorption capacities of 491.3 mg g^−1^ (MB, pH = 10) and 613.5 mg g^−1^ (EBT, pH = 1), according to the Langmuir model, which further confirmed that the 2.0 PAN/GO/PDA composite membrane was a promising adsorbent of MB and EBT (Appendix A).

This adsorption-desorption property indicates that the PAN/GO/PDA membranes possess desirable antifouling abilities and reusability. Through the adsorption in alkaline solutions and desorption in acidic solutions for MB, or the adsorption in acidic solutions and desorption in alkaline solutions for EBT, the reusability of 2.0 PAN/GO/PDA has been measured. As shown in Figure 5f, after the 10th absorption-desorption cycle, about 73.8% and 53.4% adsorption capacities for MB and EBT molecules were maintained by the 2.0 PAN/GO/PDA membranes. This indicates an appreciable reusability of the PAN/GO/PDA membranes in pollutant removal. In addition, the adsorption experiments with the membranes were completed under the drive of the solution’s own gravity, without external power. Compared with power-driven adsorption materials, the membrane material is more energy-efficient and does not require complex engineering equipment, so it has obvious advantages for the purification of outdoor natural water bodies.

### 3.4. Adsorption Mechanism of the PAN/GO/PDA Membrane

The hydrophilicity of the PAN/GO/PDA membrane, with long-time water retention, is a requisite for the adequate adsorption of dye molecules and heavy metal ions. As shown in Figure 6a, the PDA can firmly link to GO sheets through π-π conjugation, forming GO/PDA sheets. The sheet structure of GO/PDA serves to bridge the distances between PAN nanofibers, only leaving some narrow slits. Due to the numerous -OH and -COOH groups on the GO/PDA sheets, the PAN/GO/PDA membrane has a favorable hydrophilicity profile, which facilitates the rapid permeation of aqueous solutions under low pressure. Since the filter path is divided into many small units by the GO/PDA sheets, the solution residence time is stretched, increasing the efficiency of the adsorption of pollutant molecules.

The pH-dependent adsorption–desorption behaviors towards MB and EBT exhibited by the PAN/GO/PDA membrane are attributed to its protonation and deprotonation effects [39]. In a neutral solution (pH = 7), both cationic MB and anionic EBT molecules can be adsorbed onto the surface of the PAN/GO/PDA membrane through π-π stacking. When the solution is changed for an alkaline solution (pH = 10), the -OH and -COOH groups from GO and PDA can be deprotonated, forming negative charge centers (-O^−^ and -COO^−^), while the MB molecule possesses two aromatic structures with positively charged sulphur atoms (S^+^); thus, an electrostatic interaction can be produced between the PAN/GO/PDA membrane and MB molecules. This electrostatic force pushes MB molecules closer to the PAN/GO/PDA membrane, the two being eventually joined by π-π conjugation. Although EBT molecules also have aromatic structures, their negatively charged -SO_3_^−^ groups repel the negative charge surrounding the PAN/GO/PDA membrane. Thus, EBT molecules cannot be adsorbed onto the surface of the PAN/GO/PDA membrane. In acidic solution (pH = 1), the amino groups of PDA are protonated, with positive charges (-NH_2_^+^) [40,41,42]. As a result, the positively charged MB molecules cannot be adsorbed in acidic solution, yet the negatively charged EBT molecules can be adsorbed onto the surface of the PAN/GO/PDA membrane. As a result, through pH adjustment, adsorption–desorption behavior can be obtained with the PAN/GO/PDA membrane in relation to pollutant molecules [39,43].

### 3.5. Mechanical Evaluation of the PAN/GO/PDA Membrane

In practical industrial applications, the stabilities and durabilities of membranes are important indexes of quality evaluation. Compared with the PAN and PAN/PDA membranes, the tensile strength and Young’s modulus profiles of the PAN/GO/PDA membrane displayed some marked advantages, the tensile strengths attained ranging from 10.7 (0.25 PAN/GO/PDA) to 16.8 MPa (2.0 PAN/GO/PDA; Figure 7a) and the Young’s modulus values ranging from 170.2 (0.25 PAN/GO/PDA) to 232.1 MPa (2.0 PAN/GO/PDA; Figure 7b) for varying proportions of materials. Appendix A lists the mechanical properties of some typically reported electrospinning membranes for comparison with the results of this work. Obviously, compared with raw PAN and some similar modified electrospinning membranes, the 2.0 PAN/GO/PDA membrane exhibits the highest mechanical properties. Because the GO sheets are of mechanical strength and can wrap fibers in the membrane, with the increase of GO ratio, the membrane structure would be more integrated and compact. Furthermore, satisfactory mechanical damage resistance was also observed for the 2.0 PAN/GO/PDA membrane. After 10 min of rapid stirring (Figure 7c), the raw PAN membrane became loose but the 2.0 PAN/GO/PDA membrane was still intact.

## 4. Conclusions

In conclusion, composite membranes of PAN/GO/PDA were fabricated and demonstrated desirable adsorption properties with respect to both dye molecules and heavy metal ions in wastewater. In addition, due to the protonation and deprotonation effects of the PAN/GO/PDA composite in acidic and alkaline solutions, respectively, positively or negatively charged centers can be formed on the surface of the membrane. This property endows the PAN/GO/PDA membrane with preferable capabilities with respect to the adsorption of anionic dye molecules and the desorption of cationic dye molecules in acidic solutions, as well as the selective adsorption of cationic dye molecules and the desorption of anionic dye molecules in alkaline solutions. The adsorption capacity ensures the removal of pollutants, while the desorption feature makes the membrane suitable for anti-pollution purposes and reusable. Benefitting from these pH-value dependent features, the PAN/GO/PDA membrane maintains an appreciable adsorption capacity after ten absorption–desorption cycles. Furthermore, the mechanical stability of this PAN/GO/PDA membrane exceeds that of other, similar reported products. This work looks forward to driving the application of membrane technology in practical sewage treatment.

## Figures and Tables

**Figure 1 membranes-12-00938-f001:**
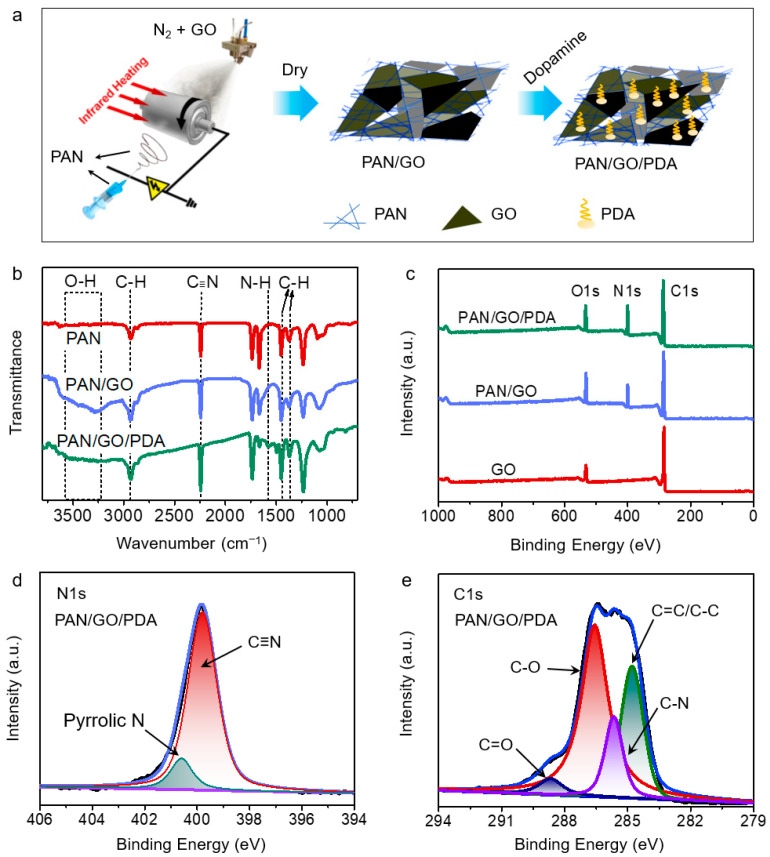
(**a**) Scheme of the synthetic route for the PAN/GO/PDA membrane. (**b**) ATR-FTIR spectra for the PAN, PAN/GO, and PAN/GO/PDA membranes. (**c**) XPS survey spectra of the PAN, PAN/GO, and PAN/GO/PDA membranes. (**d**) High-resolution image of N1s and (**e**) C1s XPS spectra for the PAN/GO/PDA membrane.

**Figure 2 membranes-12-00938-f002:**
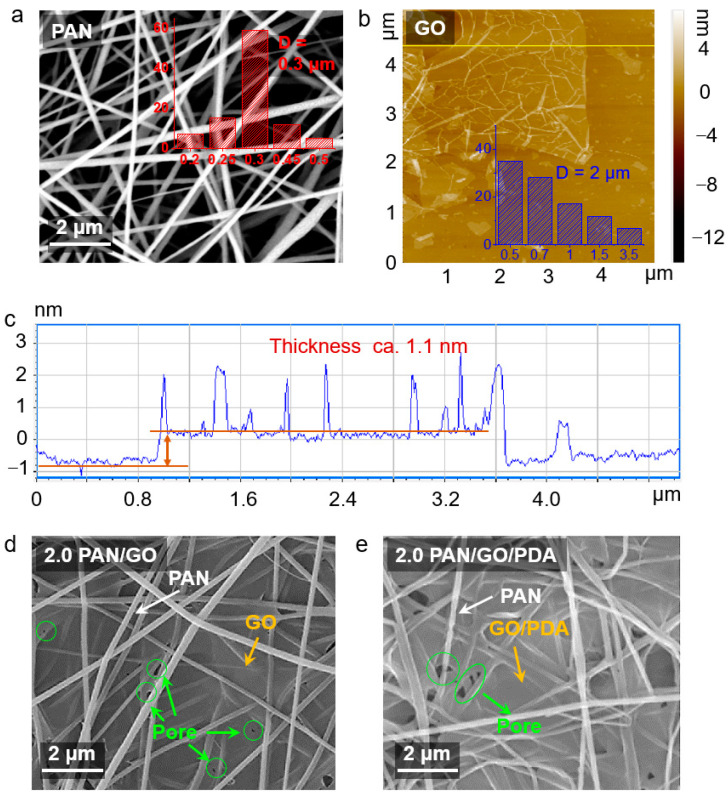
(**a**) SEM image of the PAN membrane. (**b**) AFM image of the GO sheets dispersed on silicon. (**c**) AFM height profile of the GO sheets. (**d**) SEM images of the 2.0 PAN/GO and (**e**) 2.0 PAN/GO/PDA membranes.

**Figure 3 membranes-12-00938-f003:**
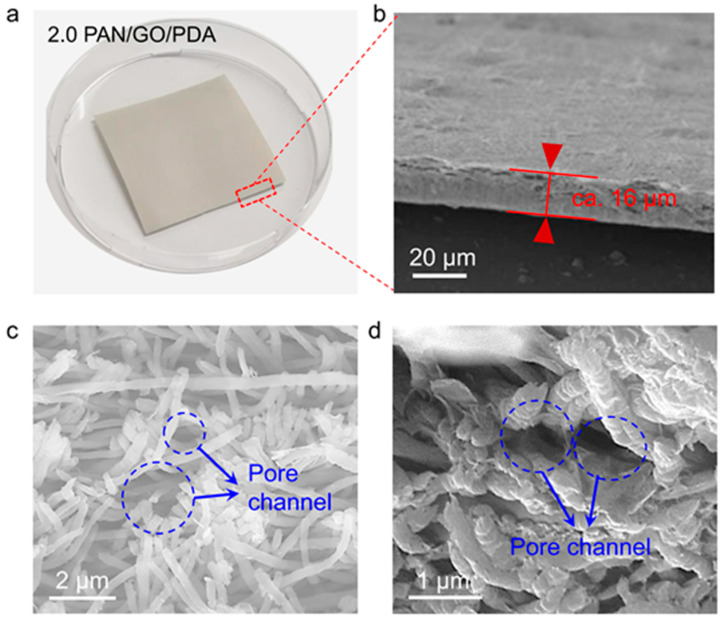
(**a**) Optical photograph of the 2.0 PAN/GO/PDA membrane. (**b**–**d**) Cross-sectional SEM images of the 2.0 PAN/GO/PDA membrane. Scale bars: (**b**) 20 μm; (**c**) 2 μm; (**d**) 1 μm.

**Figure 4 membranes-12-00938-f004:**
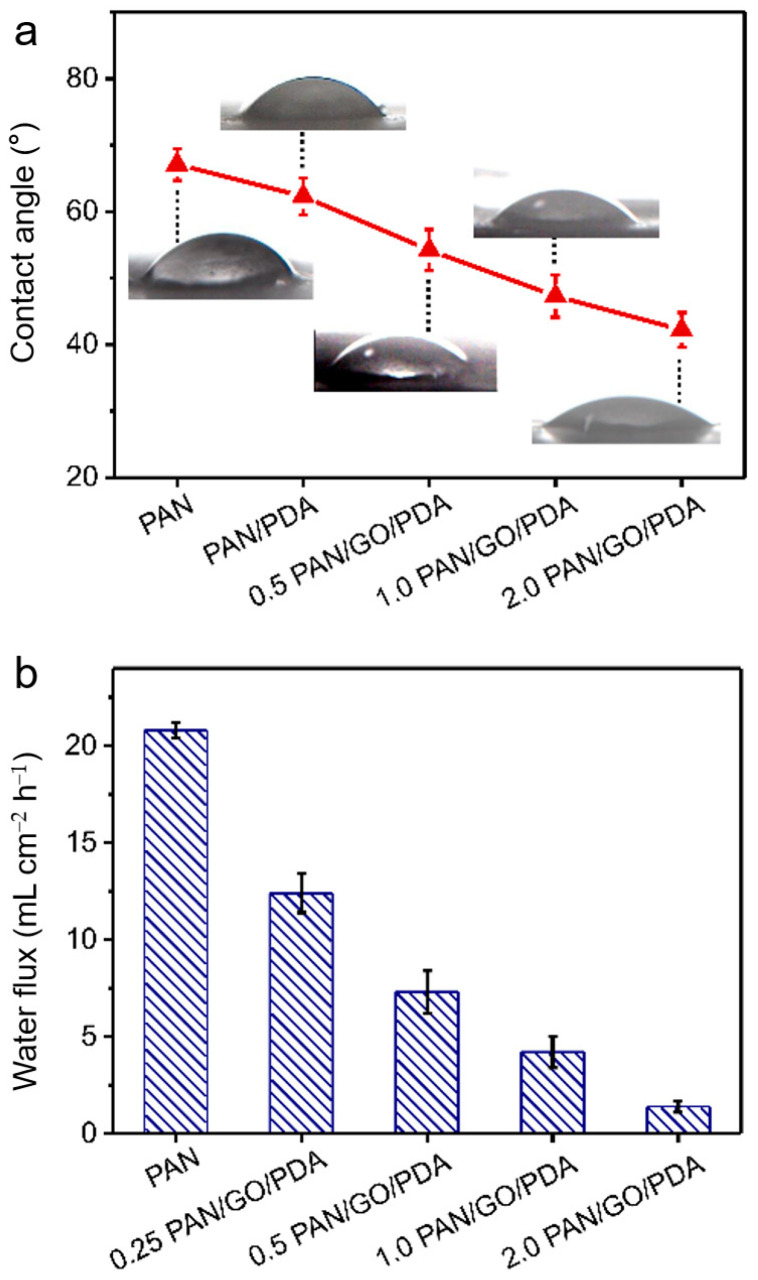
(**a**) Water contact angles of PAN, PAN/PDA, 0.5 PAN/GO/PDA, 1.0 PAN/GO/PDA, and 2.0 PAN/GO/PDA. (**b**) Pure water fluxes of PAN/PDA, 0.25 PAN/GO/PDA, 0.5 PAN/GO/PDA, 1.0 PAN/GO/PDA, and 2.0 PAN/GO/PDA.

**Figure 5 membranes-12-00938-f005:**
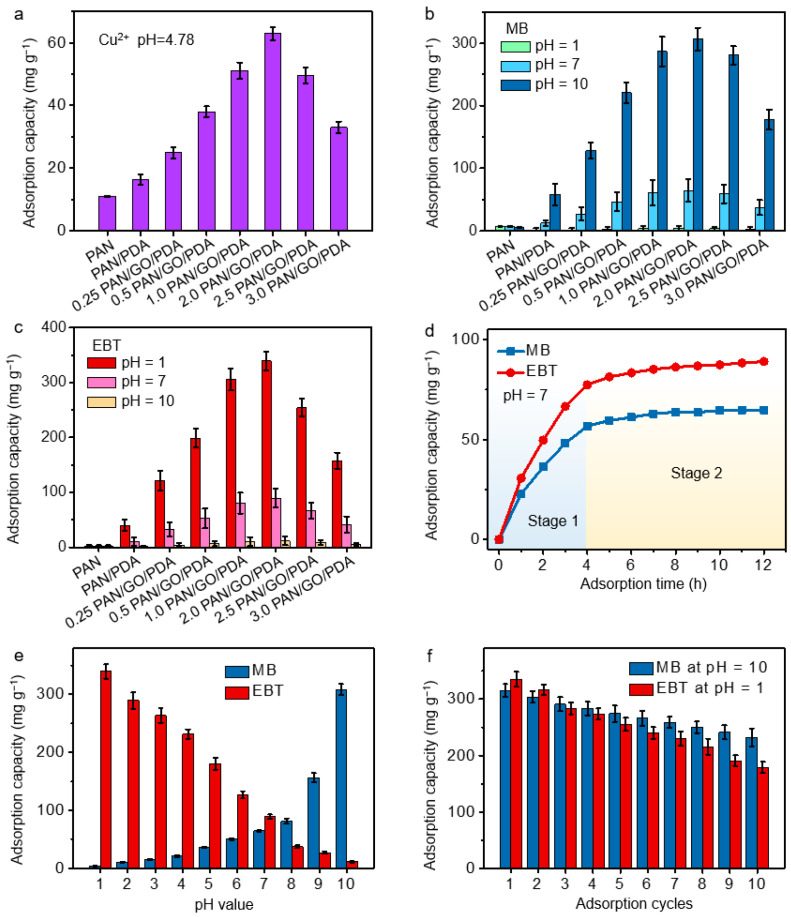
(**a**) Contrast of the experiments examining the membranes’ Cu^2+^ ion adsorption capacities (pH = 4.78). (**b**) Contrast of the experiments examining the membranes’ MB and (**c**) EBT adsorption capacities at pH = 1, 7, and 10. (**d**) The adsorption curves of MB and EBT for 2.0 PAN/GO/PDA with change over time at pH = 7. (**e**) The pH-dependent MB and EBT adsorption performances for 2.0 PAN/GO/PDA. (**f**) The reusability tests for 2.0 PAN/GO/PDA in solutions of MB (pH = 10) and EBT (pH = 1).

**Figure 6 membranes-12-00938-f006:**
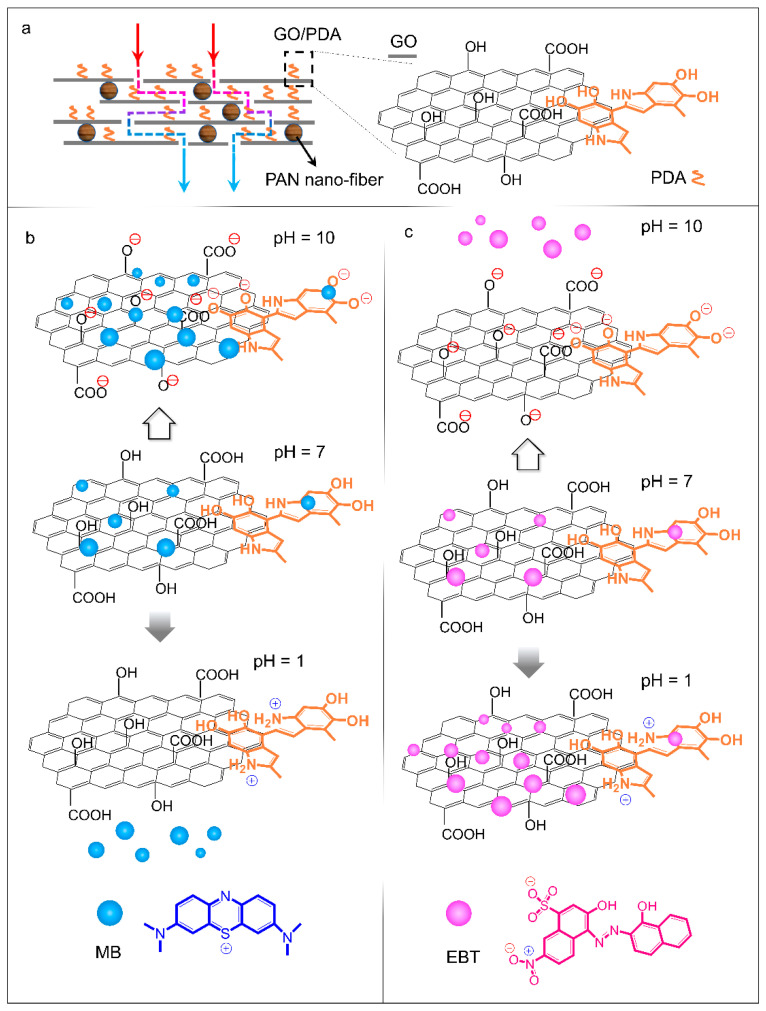
(**a**) Illustration of the construction of the PAN/GO/PDA membrane and the water-purification process. (**b**) Illustration of the pH-dependent adsorption–desorption mechanism of the PAN/GO/PDA membrane for cationic MB and (**c**) anionic EBT molecules.

**Figure 7 membranes-12-00938-f007:**
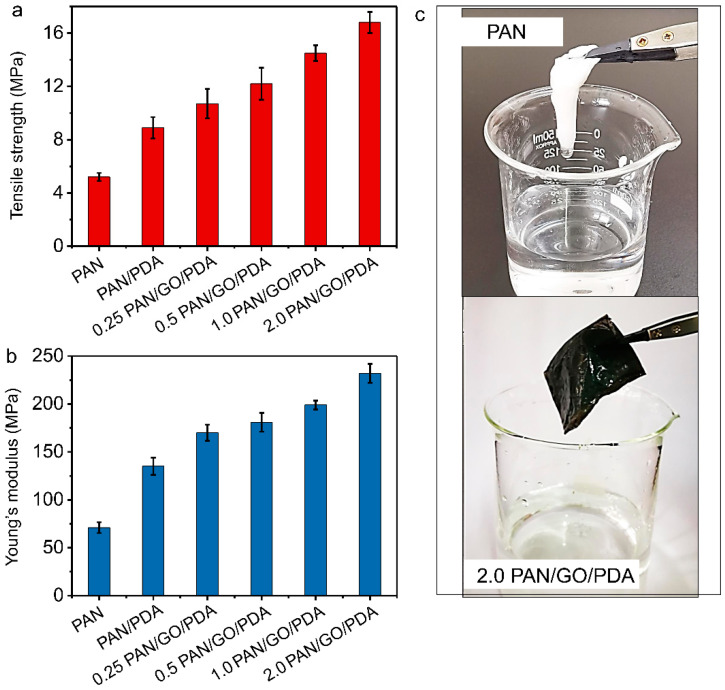
(**a**) Tensile strengths and (**b**) Young’s moduli of the composite membranes. (**c**) Comparison of the PAN and 2.0 PAN/GO/PDA membranes after being dipped and stirred (10 min) in water.

## Data Availability

The data presented in this study are available on request from the corresponding author.

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
