# Peer review of "Recyclable Composite Membrane of Polydopamine and Graphene Oxide-Modified Polyacrylonitrile for Organic Dye Molecule and Heavy Metal Ion Removal"

_membranes, 2022, doi:10.3390/membranes12100938_

Round 1
Reviewer 1 Report
1. How did you confirm your conclusion that this composite membrane displays the highest mechanical stability compared with other reported similar membranes?
2. Please remove ‘0. How to Use This Template’
3. Line 232-234, ‘…a single-layered structure with ca. 2 nm thichness…’, the thickness of a single layer GO should be less than 2 nm, right? Please refer to the literature indicating the thickness of monolayer graphene oxide.
4. Line 234-247, please cite literature to explain why spray rate affects GO coverage.
5. The explanation of this paper is not sufficient. In the results discussion part, many of them are the description of the results without the reason analysis.
6. There are few references cited to support the reliability of the discussion, please add accurate references to support the analysis.
7. In this paper, a dead-end filtration setup is used in the static adsorption membrane, which is usually used in the ultrafiltration, nanofiltration or reverse osmosis process, and depends on the size screening and charge effect can be very well removal of dyes and heavy metals. What are the advantages of the adsorption membranes compared with the pressure driven membranes? can the membranes in this paper industrialize?
In this paper, PAN/GO/PDA composite adsorption membranes were prepared by electrostatic spinning and PDA modification. The prepared membrane has good adsorption performance, but there are some problems such as insufficient explanation and not highlighting the advantages of engineering application. In particular, the dead-end setup is used in this work, but the adsorption membrane here seems to be less practical compared to the nanofiltration
Reviewer 2 Report
This manuscript describes the synthesis and the characterization of composite membranes made of the association of polydopamine, graphene oxide and polyacrylonitrile. An important part of the manuscript focuses on the synthesis of the material and the characterization (both physical and chemical). The material is applied to the sorption of copper (as a model for metal recovery) and to an anionic dye and a cationic dye. This section is less documented and would probably deserve more attention. The Authors could consider the following comments:
(1) The editing of the manuscript would require special attention; for example:
- Section 0 should be removed (“how to use this template”),
- Line 49,
- space between numbers and relevant units,
- interline spacing
- editing of references should be homogenous (for example the use of capitals in reporting the titles)
- manage the use of the capitals in the core of the sentences (Pages 91-92, for example
- apparently, the lines 120-121 are not located at the right place (case of polydopamine-functionalization placed in the section simply focused on PAN membrane and PAN/GO nanofiber membrane),
- some items would deserve appropriate reference (for example Hummer’s method, at Line 96).
(2) How is managed the “stacking in cross” while using the roller system to effectively produce “dense” membranes (as appearing in Figure 3 a/b)? This would probably help the non-specialist reader. This corresponds to questioning the naming of the material with the term membrane. Apparently, the authors spin the fibers on a roller but the method for assembling the membrane is not clear enough. Please could you elaborate more on this point?
(3) Some complementary characterizations would probably enrich the discussion of some assumptions. For example, analyzing the textural characteristics of the different membranes would help in supporting the conclusions on porous properties (SEM images are probably not sufficient).
(4) The Authors briefly comment on the study of hydrophilic behavior of the materials. This is interesting. However, a more detailed discussion would probably useful for discussing and explaining the atypical behavior of the 0.25 PAN/GO/PDA sample (Figure S2).
(5) For the study of sorption properties, a more detailed description of experimental procedures (and more specifically experimental conditions) would be useful, at least for the case of copper.
(6) The Authors selected pH 7 as the pH for testing copper recovery. Why? Clarifying this point would be welcome. Since, the effective concentration of the metal ion is not reported; it is difficult evaluating the possible occurrence of precipitation phenomenon.
(7) Concerning the effect of the pH on the sorption (and the interpretation of sorption mechanisms), the determination of the PZC value of the sorbents would be welcome. This information would be useful for supporting (and justifying) the conclusions raised by the Authors. This information could be completed by the pKa values of the reactive groups on the target dyes.
(8) Though I understand that the main objective of the paper focuses on the synthesis and characterization of materials, some complementary information relative to sorption process would be useful; for example, the sorption isotherms would bring complementary information and an effective tool for comparing these materials with alternative materials.
(9) Comparing the sorption properties of the materials with comparative materials (in the literature) would enrich the perspectives of this work (as the Authors did for mechanical properties (supposed Table S2).
(10) XPS peak positions. Frequently, the system is calibrated with a C 1s signal (Cadv., and other low BE signals associated with C=C and C-C) at 284.8 eV. Herein, the C 1s signal is set at 285.3 eV. This may affect the positioning (eV) of other typical signals (and give difficulty to compare to literature and assignments).
Round 2
Reviewer 1 Report
The author has responded well to all the questions from reviewers, and the manuscript has met the requirements for publication.